# The dynamic properties of a nuclear coactivator binding domain are evolutionarily conserved

Elin Karlsson [1], Frieda A. Sorgenfrei [1,3], Eva Andersson[1], Jakob Dogan[1], Per Jemth [1✉] & Celestine N. Chi [1,2✉]

Evolution of proteins is constrained by their structure and function. While there is a consensus that the plasticity of intrinsically disordered proteins relaxes the structural constraints on evolution there is a paucity of data on the molecular details of these processes. The Nuclear Coactivator Binding Domain (NCBD) from CREB-binding protein is a protein interaction domain, which contains a hydrophobic core but is not behaving as a typical globular domain, and has been described as 'molten-globule like'. The highly dynamic properties of NCBD makes it an interesting model system for evolutionary structure-function investigation of intrinsically disordered proteins. We have here compared the structure and biophysical properties of an ancient version of NCBD present in a bilaterian animal ancestor living around 600 million years ago with extant human NCBD. Using a combination of NMR spectroscopy, circular dichroism and kinetics we show that although NCBD has increased its thermodynamic stability, it has retained its dynamic biophysical properties in the ligand-free state in the evolutionary lineage leading from the last common bilaterian ancestor to humans. Our findings suggest that the dynamic properties of NCBD have been maintained by purifying selection and thus are important for its function, which includes mediating several distinct protein-protein interactions.

[1] Department of Medical Biochemistry and Microbiology, Uppsala University, BMC Box 582, SE-75123 Uppsala, Sweden. [2] Department of Pharmaceutical Biosciences, Uppsala University, BMC Box 582, SE-75123 Uppsala, Sweden. [3] Present address: acib GmbH, Krenngasse 37, 8010 Graz c/o University of Graz, Institute of Chemistry, NAWI Graz, BioTechMed Graz, Heinrichstrasse 28, 8010 Graz, Austria. ✉email: Per.Jemth@imbim.uu.se; Celestine.Chi@imbim.uu.se

Evolution has shaped proteins into a wide spectrum of structure, stability, and dynamics, with fully disordered proteins at one end of the scale and well-folded globular proteins at the other one[1]. It is becoming clear that dynamic properties per se are important for protein function[1–3] but it is not trivial to prove that the dynamics are essential for biological function and not merely a general property of proteins. The nuclear coactivator binding domain (NCBD) is a small (~50 residues) domain from CREB-binding protein (CBP, also called CREBBP), which is a transcriptional coactivator with histone acetylase activity and is present in all animals[4]. The NCBD domain has a hydrophobic core, but is very dynamic and even verging on being intrinsically disordered[5–7]. It has therefore been described as "molten-globule-like"[8–11]. The fact that NCBD possesses properties of both intrinsically disordered and globular proteins makes it an interesting system for assessing the role of dynamics in protein structure and how it modulates function. We have previously subjected NCBD to "evolutionary biochemistry"[12,13]. Thus, we resurrected and characterized ancestral versions of NCBD and its protein-ligand CBP-Interacting Domain (CID) from the NCOA1, 2, and 3 protein family (with three members in human called Src1, Tif2, and ACTR, respectively) to address the evolution of affinity in this protein–protein interaction. The oldest "maximum likelihood" (ML) NCBD variant we could resurrect was present before the Cambrian period, some 540–600 million years ago, in the common ancestor of all present-day animals with bilaterian symmetry. These animals are divided into deuterostomes and protostomes (D/P) and the ancestral NCBD domain is denoted $NCBD_{D/P}^{ML}$. It was shown to bind the CID domain with relatively low affinity ($K_d \sim$1–5 μM) whereas more recent variants of NCBD in the vertebrate lineage (ca. 440 million years ago) had acquired a present-day affinity for CID ($K_d \sim$0.1–0.2 μM). Structural characterization of ancestral and present-day CID/NCBD complexes showed that the increase in affinity observed for more modern NCBD variants was due to a combination of factors. These include several new interactions driving structural and dynamic changes in the complex, such as the formation of a third α helix in CID upon interaction[13]. In the present paper, we investigate the structural and biophysical properties of the ligand-free state of the ancestral Cambrian-like low-affinity $NCBD_{D/P}^{ML}$ and compare it to that of high-affinity present-day $NCBD_{Human}$ to track the evolution of structure and dynamics and their relation to function. The comparison reveals that the overall properties of the ancestral $NCBD_{D/P}^{ML}$ domain are very similar to those of the present-day $NCBD_{Human}$. In general, conservation implies function. Thus, this conservation of a highly dynamic structure and molten-globule-like properties suggest that these features are subject to purifying selection and hence that dynamics are indeed important for the biological function of NCBD.

## Results

**The structures of ancient $NCBD_{D/P}$ and extant $NCBD_{Human}$ are similar but not identical**. Because of its dynamic properties, it is challenging to solve the structure of ligand-free NCBD[8]. First, we performed experiments with $^{15}$N labeled $NCBD_{D/P}^{ML}$. We discovered by serendipity that low pH significantly improved the quality of the $^{1}$H$^{15}$N heteronuclear single quantum coherence (HSQC) spectrum resulting in well-dispersed peaks for $NCBD_{D/P}^{ML}$ (Fig. 1), and allowing for a near-complete assignment of the backbone and side-chain residues from NMR 3D experiments using $^{13}$C and $^{15}$N labeled samples. The NMR structure was thus determined at pH 2.4 by measuring 3D 1H-1H $^{13}$C/$^{15}$N-resolved NOESY experiments for distance restraints determination and $^{3}J_{HNHA}$ for

dihedral angle determination. We determined a structural ensemble consisting of 20 structures. A minimum of one long-range and 2–3 medium-range NOEs were used per residue to determine the distance restraints, far fewer than the average for folded domain proteins, but sufficient to define the overall global fold. It is worth noting that the low number of NOEs might be the reason we observe differences between the NCBD variants as detailed below. This is also the reason we searched for conditions (different pH) in our NMR structure analyses that gave minimal conformational heterogeneity, thus enough NOE distance restraints. However, it should be noted that four residues (Ser2079, Gln2082, Ala2098, and Gln2109) could not be resolved under any of the conditions tested, limiting the structural information that could be obtained from these regions.

The structure of $NCBD_{D/P}^{ML}$ was found to be overall consistent with that of the free[8] and bound $NCBD_{Human}$[6] and bound $NCBD_{D/P}^{ML}$ determined at pH 6.8[13] (Figs. 1, 2). However, a detailed comparison between $NCBD_{D/P}^{ML}$ and $NCBD_{Human}$ shows that the orientation of helix 1 (α1) and helix 3 (α3) differ between the structures. Upon structural alignment of $NCBD_{Human}$ and $NCBD_{D/P}^{ML}$, we observed that both the α1 and α3 from $NCBD_{D/P}^{ML}$ are slightly tilted away relative to that of $NCBD_{Human}$ (Fig. 1). On the other hand, as shown previously[13], CID-bound $NCBD_{D/P}^{ML}$ and $NCBD_{Human}$ are structurally more similar (Fig. 2).

The amino acid sequence of $NCBD_{D/P}^{ML}$ was reconstructed using phylogenetic methods[12]. The probability is low that the resulting maximum likelihood sequence is identical to the actual sequence present in the ancestor. Thus, $NCBD_{D/P}^{ML}$ is rather one of a large number of likely ancestral variants with similar properties. We wondered if errors in the sequence of $NCBD_{D/P}^{ML}$ resulted in the differences we observe in helix α1 and α3 compared to $NCBD_{Human}$. To investigate this further, and to test how structurally robust the dynamic NCBD domain is, we performed a control experiment where we expressed, purified, and characterized an alternative $NCBD_{D/P}$ variant denoted $NCBD_{D/P}^{AltAll}$. In this $NCBD_{D/P}^{AltAll}$ variant, all residues with a posterior probability lower than around 0.9[12] were replaced with the second most likely residue at that position. For example, residue 2107 is His with 87% probability and Gln with 12% probability in the ancestral $NCBD_{D/P}$. Thus, $NCBD_{D/P}^{ML}$ has a His in position 2107 and $NCBD_{D/P}^{AltAll}$ a Gln residue. $NCBD_{D/P}^{AltAll}$ can be considered a "worst-case scenario" variant and good control of the robustness of conclusions drawn from resurrection experiments. If ML and AltAll variants display similar properties it is very likely that the actual ancestral protein shares these properties as well. Because of this, an AltAll variant is a convenient alternative to combinations of point mutations[14]. $NCBD_{D/P}^{AltAll}$ differs from $NCBD_{D/P}^{ML}$ by amino acid substitutions at six positions (Fig. 1c). Three of the differences between $NCBD_{D/P}^{ML}$ and $NCBD_{D/P}^{AltAll}$ are in the N-terminus, but only one of these is in a structured region, a Gln2065→Pro at the beginning of α1. The other three differences are Ser2078→Asn in the loop between α1 and α2, a solvent-exposed Gln2088→His in α2, and His2108→Gln at the C-terminus. We solved the NMR structure of $NCBD_{D/P}^{AltAll}$ (Fig. S1) and found that it folds into a similar structure as the other NCBD domains. The overall RMSD between $NCBD_{D/P}^{AltAll}$ and $NCBD_{D/P}^{ML}$ was 7.9 Å. As a comparison, the overall RMSD between $NCBD_{Human}$ and $NCBD_{D/P}^{AltAll}$ or $NCBD_{D/P}^{ML}$ were 4.5 and 7.0 Å, respectively.

Detailed inspection of the structures revealed differences in the backbone between $NCBD_{D/P}^{ML}$ and $NCBD_{D/P}^{AltAll}$, especially in the orientations of the helices. An alignment of the structures of $NCBD_{D/P}^{AltAll}$, $NCBD_{D/P}^{ML}$, and $NCBD_{Human}$, shows differences between α1, α2, and α3 (Fig. 1, Fig. S1, and Table S1). One reason for the differences is that there are several long-range NOEs in $NCBD_{D/P}^{AltAll}$ that are not present in $NCBD_{D/P}^{ML}$, between the

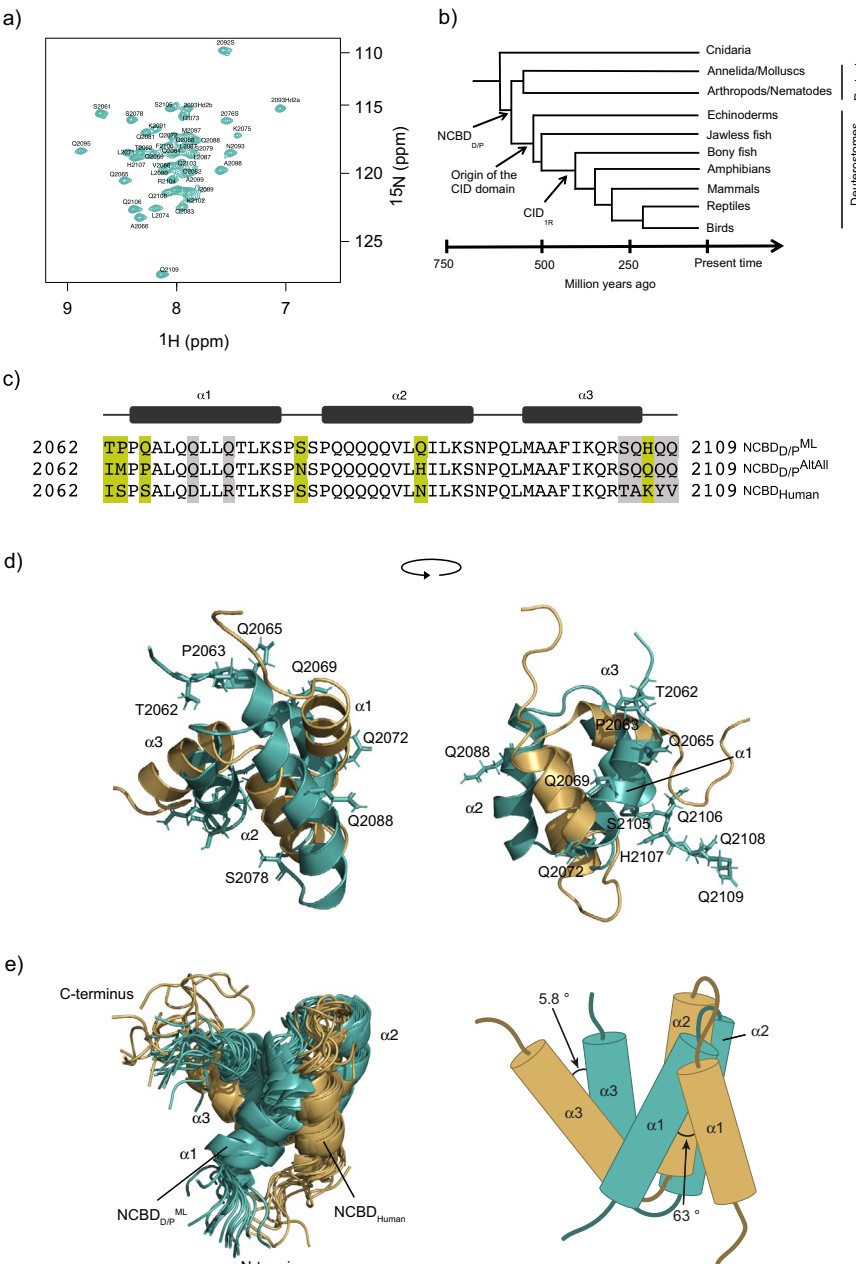

**Fig. 1 Structures of ancestral and human NCBD domains. a** $^{1}$H-$^{15}$N HSQC spectrum of NCBD$_{D/P}^{ML}$ at pH 2.4 with assigned residues. All assignments and the structural coordinates have been deposited in the PDB data bank with accession number BMRB ID 34635 and PDB code: 7OSR. **b** A schematic tree depicting the relationship between the deuterostome/protostome (D/P) ancestor and modern species. The time points on the x-axis are approximate, in particular the divergence of protostomes and deuterostomes. **c** Sequence alignment for NCBD$_{Human}$, NCBD$_{D/P}^{ML}$, and NCBD$_{D/P}^{AltAll}$. The residues differing between NCBD$_{D/P}^{ML}$ and NCBD$_{D/P}^{AltAll}$ are marked in celery green while additional differences to NCBD$_{Human}$ are marked in gray. **d, e** Comparison of the NMR structure of ancestral NCBD$_{D/P}^{ML}$ solved at pH 2.4 (teal, PDB code: 7OSR), and the previously determined human NCBD (gold, 2KKJ)[8]. Note that the structure of human NCBD was solved using a slightly longer construct and at pH 6.5. **d** Structural alignment of NCBD$_{D/P}^{ML}$ (teal) and NCBD$_{Human}$ (gold) with the residues that differ between the two variants represented as stick models. **e** Structural alignment of the 20 lowest energy structures of the NCBD$_{D/P}^{ML}$ domain with those of NCBD$_{Human}$. Tilt angles for α1 and α3 from the lowest energy structures. See Table S1 for the determination of the tilt angles.

following pairs of residues: 2068/2082, 2069/2080, 2070/2083, and 2071/2099. There is also an NOE between Ile2062 Hγ and Pro2065 Hα, two of the substituted residues in NCBD$_{D/P}^{AltAll}$.

There appears to be a consistency in the hydrophobic core of both NCBD$_{D/P}^{ML}$ and NCBD$_{D/P}^{AltAll}$ since the single Phe residue at position 2100 is packed and buried in a similar fashion in the two variants. Phe2100 is stabilized by Leu2090 in NCBD$_{D/P}^{ML}$, and both Ile2089 and Leu2090 in NCBD$_{D/P}^{AltAll}$ (Fig. 3). Interestingly, this is the same kind of packing that was observed in the free NCBD$_{Human}$ domain[8], suggesting conserved features in the hydrophobic core. We also compared the structures by calculating the difference in solvent accessible surface area (ΔSASA) between folded and unfolded proteins using ProtSA (Table S1). We obtained values of ΔSASA of 1264, 1795, and 1825 Å$^{2}$ for NCBD$_{D/P}^{ML}$, NCBD$_{D/P}^{AltAll}$, and NCBD$_{Human}$, respectively, indicating an overall similar burial of residues for NCBD$_{D/P}^{AltAll}$, and NCBD$_{Human}$. The lower ΔSASA for

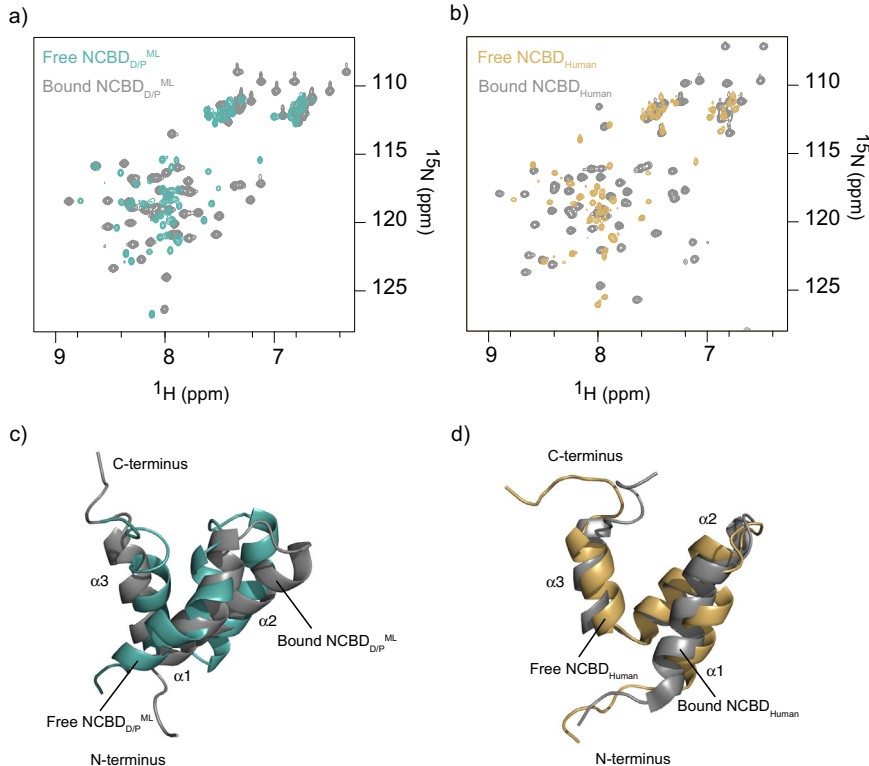

**Fig. 2 Comparison of the ligand-free and bound ancestral NCBD$_{D/P}$$^{ML}$ and extant NCBD$_{Human}$.** The ligand-bound conformations (PDB codes 6ES5 and 6ES7)[13] and the structure of free human NCBD (PDB code: 2KKJ)[8] were previously determined. **a** Overlay of the $^1$H-$^{15}$N HSQC spectra of free NCBD$_{D/P}$$^{ML}$ at pH 2.4 (teal) and CID$_{1R}$$^{ML}$-bound NCBD$_{D/P}$$^{ML}$ at pH 6.8 (gray). **b** Overlay of the $^1$H-$^{15}$N HSQC spectra of free NCBD$_{Human}$ at pH 6.5 (gold) and CID$_{Human}$-bound NCBD$_{Human}$ (gray) at pH 6.8. **c** Structural alignment of free NCBD$_{D/P}$$^{ML}$ (teal, 7OSR, pH 2.4) and CID$_{1R}$$^{ML}$-bound NCBD$_{D/P}$$^{ML}$ (gray, 6ES5, pH 6.8). **d** Structural alignment of free NCBD$_{Human}$ (gold, 2KKJ, pH 6.5) and CID$_{Human}$-bound NCBD$_{Human}$ (gray, 6ES7, pH 6.8).

NCBD$_{D/P}$$^{ML}$ stems from a higher SASA in the folded state, i.e., a less compact folded structure. This result, which is consistent with the overall higher structural similarity between NCBD$_{D/P}$$^{AltAll}$, and NCBD$_{Human}$, may be due to the lack of NOE constraints as discussed above, but might also reflect the dynamic properties of NCBD, where small changes in sequence could modulate packing of helices. In this respect, errors in the reconstruction could affect the details of the structure.

Despite the differences discussed in the previous paragraphs, the overall biophysical properties of NCBD$_{D/P}$$^{ML}$ and NCBD$_{D/P}$$^{AltAll}$ were found to be similar (Fig. S2) corroborating the overall similarity seen with the NMR structures. Specifically, to test the affinity of NCBD$_{D/P}$$^{AltAll}$ for CID, we expressed and purified ML and AltAll versions of CID$_{1R}$, which was previously reconstructed using sequences from present-day vertebrates. Thus, CID$_{1R}$ represents the domain present in the ancestor of an early vertebrate some 440 million years ago[12]. $K_d$ values of all four combinations (NCBD$_{D/P}$$^{ML}$ and NCBD$_{D/P}$$^{AltAll}$ with CID$_{1R}$$^{ML}$ and CID$_{1R}$$^{AltAll}$) were between 2–9 µM (Fig. S2a, b), which is in the same range as previous estimates of affinity of the ancestral complex[12,13]. Furthermore, the stability as monitored by urea denaturation (Fig. S2c) and thermal denaturation (Fig. S2d, e) were similar for NCBD$_{D/P}$$^{ML}$ and NCBD$_{D/P}$$^{AltAll}$. Thus, based on the highly similar biophysical properties of NCBD$_{D/P}$$^{ML}$ and NCBD$_{D/P}$$^{AltAll}$ we infer that our conclusions are robust with regard to the uncertainty in the reconstructed sequence.

**Ancient NCBD$_{D/P}$ is less thermodynamically stable than extant NCBD$_{Human}$ but displays a similar pH dependence with regard to structure and stability.** Since the NCBD$_{D/P}$$^{ML}$ structure was

determined at low pH, we next investigated how pH influences the fold and stability. First, we performed $^1$H-$^{15}$N HSQC-monitored pH titrations of NCBD$_{D/P}$$^{ML}$ and compared them to those of NCBD$_{Human}$ at different pH values (Fig. 4). The NMR titration experiments showed that NCBD$_{D/P}$$^{ML}$ is as folded at low pH as it is at high pH values. However, the peaks ($^1$H-$^{15}$N) are much sharper (reduced line-width) at lower pH values indicating less conformational heterogeneity. It should be noted that the sharpness of NMR lines does not depend only on conformational heterogeneity but also on the rate of exchange between these conformational states. With respect to the NMR time regime, fast exchange will result in sharp peaks, intermediate exchange will result in broadening of the spectra while slow exchange will result in distinct conformational peaks. Second, the CD-monitored urea denaturation experiments showed identical stability patterns at two pH values (3.0 and 7.4) for both ancient NCBD$_{D/P}$$^{ML}$ and extant NCBD$_{Human}$ (Fig. 4c, d, Fig. S3, and Table S2). This indicates that while at lower pH the NCBD$_{D/P}$$^{ML}$ still exhibits the same stability as at high pH values, there is a shift in conformational heterogeneity towards a single state at low pH. Furthermore, whereas the apparent thermodynamic stability differs by ~1 kcal mol$^{-1}$ for ancient NCBD$_{D/P}$$^{ML}$ and extant NCBD$_{Human}$, both are largely unaffected by a drop in pH to 3.0 (Table S2), consistent with few titratable groups in this pH range (one His in NCBD$_{D/P}$$^{ML}$ and one Asp in NCBD$_{Human}$). Previous experiments suggested that the ancestral NCBD$_{D/P}$$^{ML}$ was slightly less stable than younger and extant NCBD variants as judged by urea denaturation experiments at pH 7.5[12]. However, the broad unfolding transition of NCBD resulting from its small hydrophobic core leads to a large error in the $m_{D-N}$ value, which makes it difficult to unequivocally determine the thermodynamic stability

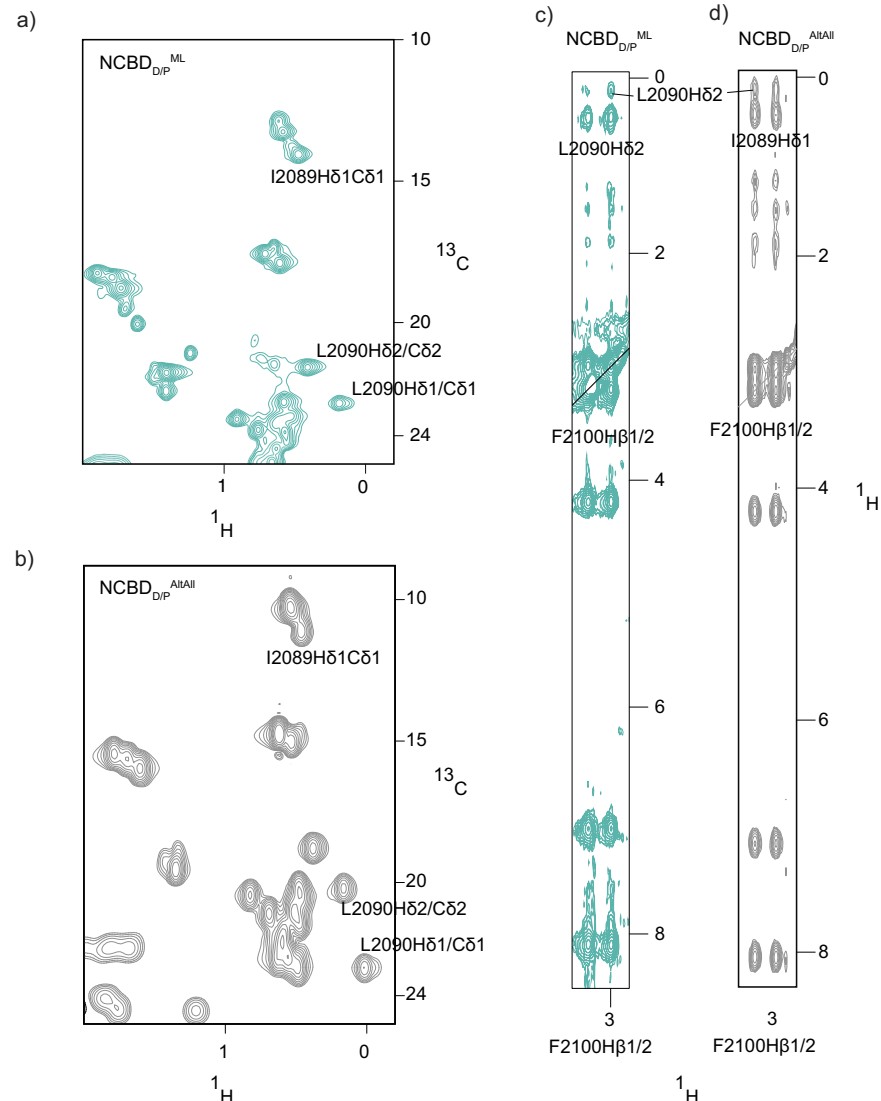

**Fig. 3 Phe2100 makes NOEs to Ile2089 and Leu2090 in NCBD$_{D/P}$$^{ML}$ and NCBD$_{D/P}$$^{AltAll}$. a** $^1$H-$^{13}$C HSQC spectra at pH 2.4 of the methyl region of NCBD$_{D/P}$$^{ML}$, showing the Hδ1-Cδ1 shifts of L2090 and I2089. **b** $^1$H-$^{13}$C HSQC spectra at pH 6.8 of the methyl region of NCBD$_{D/P}$$^{AltAll}$, showing the Hδ1-Cδ1 shifts of L2090 and I2089. **c** Strip from a $^1$H-$^1$H $^{13}$C-resolved NOESY showing NOEs from the Hβ1/2 atoms of Phe2100 and the associated NOEs to Leu2090Hδ1 for NCBD$_{D/P}$$^{ML}$. **d** Strip from a $^1$H-$^1$H $^{13}$C-resolved NOESY showing NOEs from the Hβ1/2 atoms of Phe2100 and the associated NOEs to Leu2090Hδ1 and Ile2089Hδ1 for NCBD$_{D/P}$$^{AltAll}$. These signature NOEs imply that both proteins form a hydrophobic core similar to that of NCBD$_{Human}$.

of the domain. However, all present results (including experiments on the stabilized NCBD$_{D/P}$$^{ML}$ variant NCBD$_{D/P}$$^{T2073W}$ (Table S2) agree well with the previous experiments and we can therefore conclude that the ancestral NCBD$_{D/P}$$^{ML}$ was less thermodynamically stable than evolutionarily younger variants. Interestingly, with some exceptions[15,16], ancestral reconstructions often yield proteins with higher thermostability, which correlates with thermodynamic stability, than extant proteins. This notion has been attributed either to a bias for stability in the reconstruction (an artifact) or to higher temperatures in past times[17–21], like during Cambrian[22,23]. In any case, ancestral NCBD appears to have populated the native state to a lesser extent than present-day human NCBD.

**Ancient NCBD$_{D/P}$$^{ML}$ and extant NCBD$_{Human}$ display similar temperature-dependent structural changes.** Previous experiments demonstrated that NCBD$_{Human}$ shows an apparent non-cooperative reversible unfolding behavior when subjected to increasing temperatures, as monitored by CD[6,8]. This behavior

may result from a low $\Delta H_{D-N}$ value rather than a true non-cooperative unfolding. (The urea denaturation experiments are consistent with a cooperative two-state unfolding.) We compared $^1$H-$^{15}$N HSQC and CD spectra, respectively, of NCBD$_{D/P}$$^{ML}$ and NCBD$_{Human}$ at different temperatures (Fig. 5). In agreement with the previous data, we observed that both variants unfold non-cooperatively with temperature as monitored by the increase in molar ellipticity at 222 nm suggesting loss of α-helix as the temperature is increased. However, for both NCBD$_{D/P}$$^{ML}$ and NCBD$_{Human}$ the signal is not completely lost even at 363 K (90 °C) underscoring that this is not a typical native to a highly-disordered-denatured state transition. Furthermore, $^1$H-$^{15}$N HSQC spectra of NCBD$_{D/P}$$^{ML}$ at temperatures up to 333 K (60 °C) showed well-dispersed peaks indicating a presumably globular, collapsed architecture (Fig. S4), corroborating the notion that NCBD retains structure at high temperatures.

**Rate constants for (un)folding of ancient NCBD$_{D/P}$$^{ML}$ and extant NCBD$_{Human}$.** The folding of human NCBD is complex,

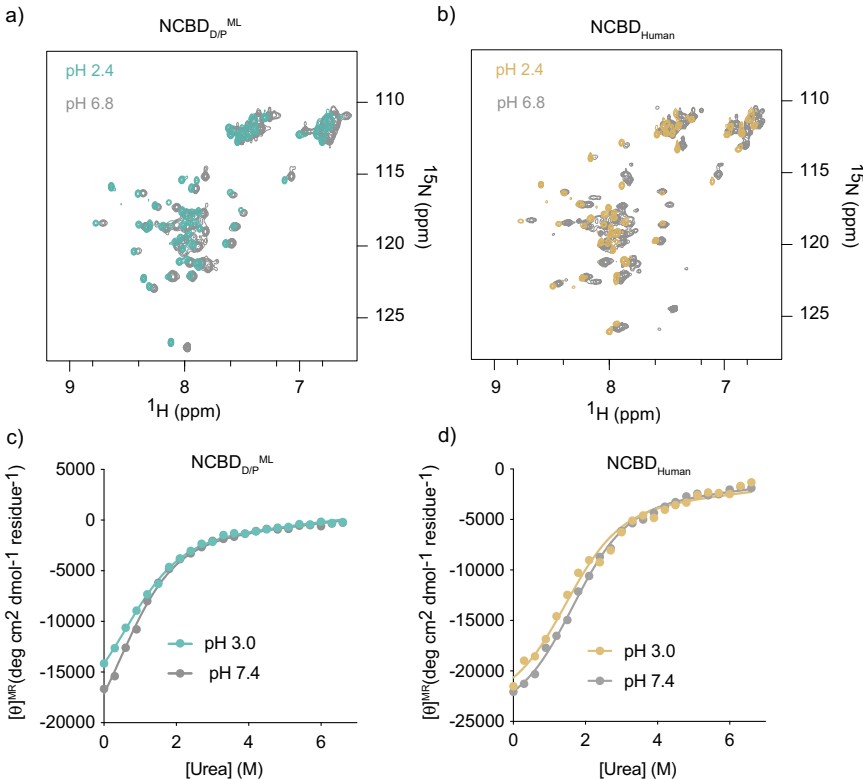

**Fig. 4 Comparison of ancestral NCBD$_{D/P}^{ML}$ and extant NCBD$_{Human}$ at two different pH values. a** Overlay of the $^1$H-$^{15}$N HSQC spectra of NCBD$_{D/P}^{ML}$ at pH 2.4 (teal) and at pH 6.8 (gray). **b** Overlay of the $^1$H-$^{15}$N HSQC spectra of NCBD$_{Human}$ at pH 2.4 (gold) and at pH 6.8 (gray). **c** Urea denaturation experiments for NCBD$_{D/P}^{ML}$ at pH 3.0 (teal) and at pH 7.4 (gray). **d** Urea denaturation experiments for NCBD$_{Human}$ at pH 3.0 (gold) and at pH 7.4 (gray).

and, based on kinetic experiments on a human NCBD variant, we proposed a three-state system involving "native" NCBD (i.e., the solved structure), an intermediate state with unknown structure and a denatured state, which likely retains significant structure under native conditions[24]. Rate constants in the same order was determined by relaxation dispersion NMR experiments by Kjaergaard et al.[10]. It is clear from the NMR experiments that low pH promotes the native state of NCBD and that pH does not affect the thermodynamic stability as monitored by urea denaturation experiments to any significant extent, neither for NCBD$_{D/P}^{ML}$ nor NCBD$_{Human}$. Previous folding experiments on NCBD$_{Human}$[24] involving jumps from low to neutral pH thus resulted in small shifts in the equilibria between the native, intermediate and denatured states yielding the observed kinetic transients. In fact, a likely explanation for the better NMR spectra at low pH is that there is less of the intermediate state under these conditions.

To further compare the biophysical properties of ancient NCBD$_{D/P}^{ML}$ and extant NCBD$_{Human}$ we performed kinetic folding experiments using stopped-flow as well as temperature-jump fluorescence spectroscopy. To be able to monitor structural rearrangements we used a variant of NCBD$_{D/P}^{ML}$ with a Trp at position 2073, previously used in kinetic binding studies. A Trp in the corresponding position was also previously used in the folding studies of NCBD$_{Human}$[24]. This Thr→Trp modification resulted in a thermodynamic stabilization of 1.2–1.4 kcal mol$^{-1}$ for both NCBD$_{D/P}^{T2073W}$ and NCBD$_{Human}^{T2073W}$ (Fig. 6a, b, Supplementary Fig. S3, and Table S2). Previous folding experiments were performed on a slightly longer NCBD$_{Human}$ variant (residues 2058–2116), which was used in the original studies of CID/NCBD[6]. In our resurrection studies, we used a shorter version of NCBD$_{Human}$, only containing the evolutionarily conserved region (residues 2062–2109). Therefore, we first repeated the

temperature-jump folding experiments for NCBD$_{Human}^{T2073W}$ and showed that the shorter construct used in our evolutionary studies displayed similar kinetics as the longer construct (Fig. 6c). We then performed folding experiments with NCBD$_{D/P}^{T2073W}$. The kinetic transients of NCBD$_{D/P}^{T2073W}$ from temperature-jump experiments had lower signal-to-noise than those of NCBD$_{Human}^{T2073W}$, but observed rate constants were in the same range (Fig. 6d). To investigate the folding kinetics further we used stopped-flow fluorimetry, which has a lower time resolution but can be used to compare folding rate constants for NCBD$_{D/P}^{T2073W}$ and NCBD$_{Human}^{T2073W}$ at low temperature (Fig. 6e, f). The stopped-flow folding experiments were conducted by making pH jumps from a 5 mM HCl solution (pH ~2.3) to a buffer solution of 20 mM sodium phosphate (pH 7.4, 150 mM NaCl, 1 M TMAO) and monitoring the relaxation through a 330 nm band-pass filter at 4 °C. The low temperature is necessary to reduce the observed rate constant such that the folding or other conformational transitions take place in the millisecond time window. We observed two kinetic phases for NCBD$_{Human}^{T2073W}$ (one fast phase with a negative amplitude and a second slower phase with a positive amplitude) but only one kinetic phase for NCBD$_{D/P}^{T2073W}$ (with a positive amplitude). The values of the observed rate constants for NCBD$_{Human}^{T2073W}$ (~300 s$^{-1}$ and ~70 s$^{-1}$, respectively) were similar to those observed previously for the longer construct[24]. The single value obtained for the NCBD$_{D/P}^{T2073W}$ was ~140 s$^{-1}$. Trp fluorescence is a very sensitive but crude structural probe. Thus, the number of kinetic phases in folding experiments could be dependent on small structural rearrangements around the Trp. The structural basis for the kinetic phases is likely shifts of conformational equilibria between two or more folded states, as mentioned above and suggested by previous studies on human NCBD[10,24,25]. Nevertheless, the observed rate constant for NCBD$_{D/P}^{T2073W}$ in the

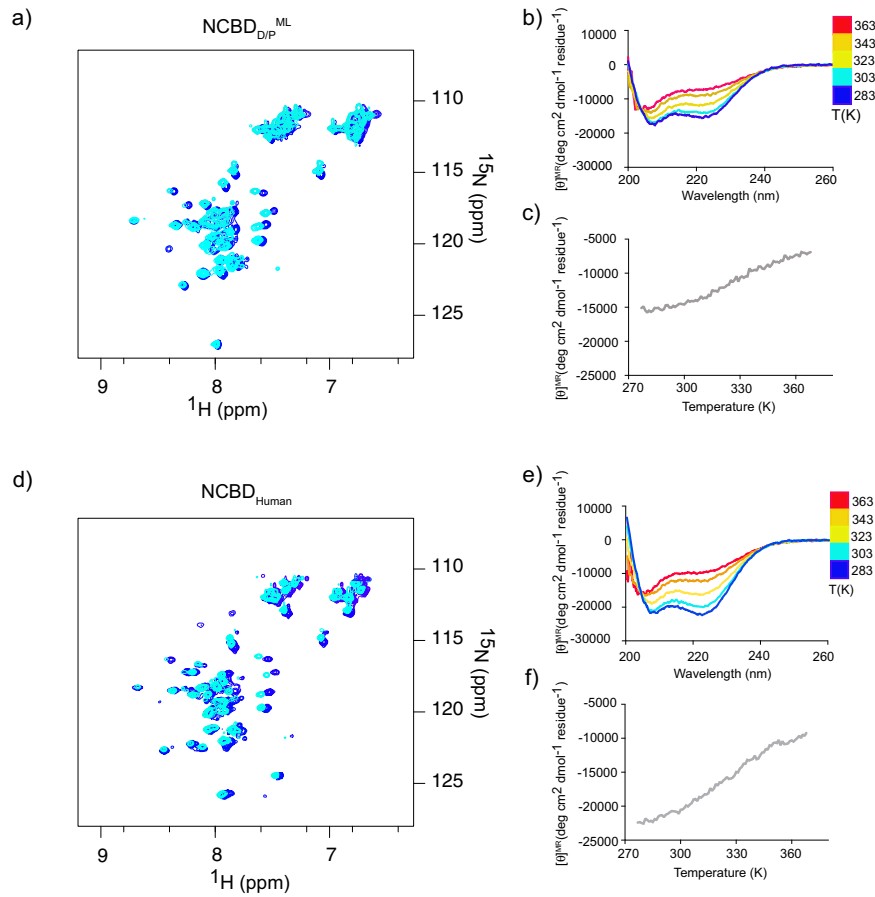

**Fig. 5 Temperature dependence for ancestral NCBD$_{D/P}$$^{ML}$ and extant NCBD$_{Human}$. a** Overlay of $^1$H-$^{15}$N HSQC spectra of NCBD$_{D/P}$$^{ML}$ at 283 K (blue) and at 303 K (cyan). **b** Circular dichroism spectra at a range of temperatures (283–363 K) for NCBD$_{D/P}$$^{ML}$. **c** Molecular ellipticity at 222 nm (probing α-helix content) as a function of temperature for NCBD$_{D/P}$$^{ML}$. All experiments in (**a–c**) were performed at pH 2.4. **d–f** Similar experiments as in panels (**a–c**), respectively, for NCBD$_{Human}$ performed at pH 6.8.

stopped-flow pH jump experiment is in the same range as the two rate constants observed under the same conditions for human NCBD.

## Discussion

The biophysics and molecular evolution of NCBD is interesting for several reasons, which all relate to the dynamic properties of NCBD. Firstly, while it has a hydrophobic core and globular shape, NCBD is a very dynamic protein domain with characteristics of an IDP[5,6,8–11,26]. Secondly, NCBD has several binding partners including the three NCOA transcriptional co-regulator paralogs[5,27], the transcription factors p53[28], Ets-2[29], and interferon factor-3[30], and viral proteins such as Tax[31] from the T-cell leukemia virus. Intriguingly, NCBD displays great conformational plasticity as shown by its complex with interferon factor-3 where the helices of NCBD adopt a distinct conformation as compared to the complex with CID[32]. Thirdly, evolutionary snapshots of the complex between CID and NCBD suggest that several new intermolecular contacts and conformational plasticity increased the affinity when going from the Cambrian-like NCBD$_{D/P}$$^{ML}$ ($K_d$ ~1–5 μM) to younger variants ($K_d$ ~0.2 μM)[12,13,33]. Fourthly, while affinity and structural order of modern NCBD/CID complexes were increased, the transition state of the folding-induced binding displayed more conformational heterogeneity as compared to the ancestral complex[34]. Our present results, how the free, unbound state of NCBD has evolved during the transition from a low-affinity to a high-affinity CID binder relate to all four points.

The most striking finding is that NCBD$_{D/P}$$^{ML}$ displays very similar molten-globule-like properties as the previously characterized human NCBD[8,10]. This conservation of properties, the borderline between a globular domain, and an IDP suggests that the dynamics of NCBD is preserved by purifying selection and therefore of functional importance. In general, structural plasticity is regarded as one main functional feature of IDPs allowing for multiple protein–protein interactions[35]. Given the high sequence identity between NCBD$_{D/P}$$^{ML}$ and human NCBD (Fig. 1), it is perhaps not surprising that the molten-globule-like properties have been conserved. Conservation of sequence is dictated by function, i.e., the affinity toward the binding partners of NCBD. Since NCBD interacts with several protein domains, both folded and disordered ones, we speculate that the dynamic nature of NCBD has been conserved to facilitate multi-partner binding. Moreover, NCBD interactions result in coupled folding and binding. Many such reactions have recently been shown to occur via a mechanism denoted templated folding, where the binding partner influences the folding pathway[36–38]. Indeed, templated folding was observed for the interaction between CID and ancestral as well as human NCBD[34,39,40]. Thus, our present findings suggest that templated folding may be a general feature of binding reactions with NCBD where the conserved dynamics facilitate sampling of different free states, heterogeneity of bound states, as well as flexibility in the pathway between free and bound states.

Going into details of the NMR model, we note that while the overall structure is robust to errors in the predicted sequence as

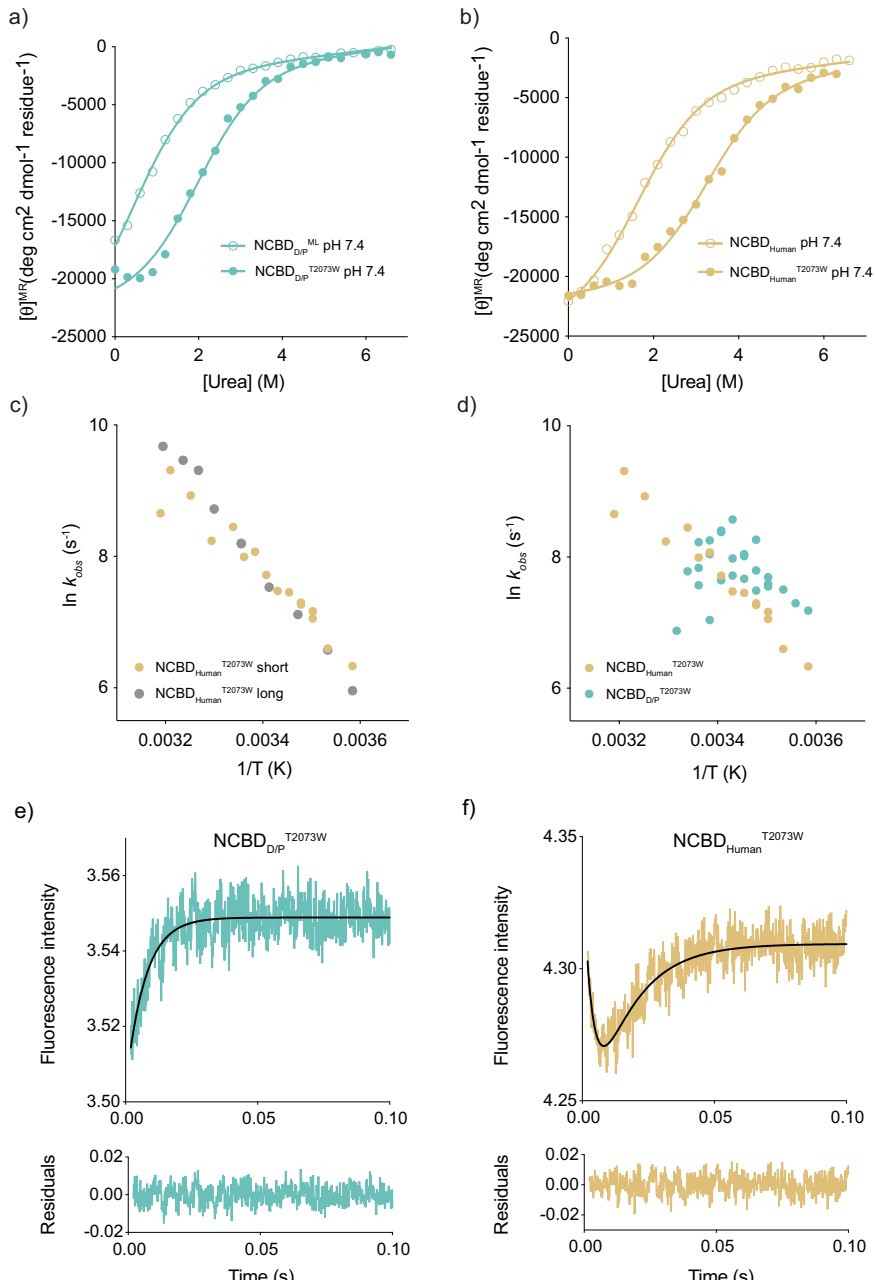

**Fig. 6 Thermodynamic stability and temperature dependence of kinetic rate constants for NCBD$_{D/P}$$^{ML}$ and extant NCBD$_{Human}$. a** Stability measurement of NCBD$_{D/P}$$^{ML}$ (open circles) and NCBD$_{D/P}$$^{T2073W}$ (filled circles) measured by urea denaturation. **b** Stability measurement of NCBD$_{Human}$$^{WT}$ (open circles) and NCBD$_{Human}$$^{T2073W}$ (filled circles). Parameters from the curve fitting are shown in Table S2. **c** Temperature jump kinetic experiments for a longer human NCBD construct used in previous studies[24] (gray) and the shorter human NCBD variant used in this study (gold). Both variants contained the T2073W mutation. **d** A comparison of observed rate constants from temperature-jump experiments for NCBD$_{Human}$$^{T2073W}$ (gold) and NCBD$_{D/P}$$^{T2073W}$ (teal). All experiments were performed in 20 mM sodium phosphate pH 7.4, 150 mM NaCl. **e, f** Kinetic (un)folding of NCBD$_{Human}$$^{T2073W}$ and NCBD$_{D/P}$$^{T2073W}$ measured by stopped-flow fluorescence spectroscopy. **e** Stopped-flow (un)folding kinetic transient of NCBD$_{D/P}$$^{T2073W}$ (teal) fitted to a single exponential function (black line). **f** (Un)folding kinetic transient of NCBD$_{Human}$$^{T2073W}$ (gold) fitted to a double exponential function (black line). Below each transient are the residuals from the respective fits. The experiments were performed by jumping from 5 mM HCl (pH ~2.3) to a final buffer concentration of 20 mM sodium phosphate (pH 7.4), 150 mM NaCl, 1 M TMAO. The relaxation was monitored using a 330 band-pass emission filter. Fitting a double exponential function resulted in $k_{obs}$ values of 290 ± 30 and 67 ± 4 s$^{-1}$ for NCBD$_{Human}$$^{T2073W}$. Fitting a single exponential function yielded a $k_{obs}$ value of 139 ± 7 s$^{-1}$ for NCBD$_{D/P}$$^{T2073W}$.

shown by the comparison of the NMR models of free NCBD$_{D/P}$$^{ML}$ and NCBD$_{D/P}$$^{AltAll}$, there are differences in for example helix orientations. Globular domains with a sequence identity as high as these NCBD variants (77–88%) would typically have identical structures due to a highly funneled energy landscape[41,42], while proteins with larger structural heterogeneity have less funneled landscapes[1]. Indeed, the differences between the structures of NCBD$_{Human}$, NCBD$_{D/P}$$^{ML}$, and NCBD$_{D/P}$$^{AltAll}$, in particular regarding the angles of the helices in relation to each other, are consistent with a less funneled energy landscape resulting in

conformational heterogeneity[11]. This heterogeneity is also consistent with the conserved dynamic properties of NCBD. In conclusion, our data suggest evolutionarily conserved dynamics of NCBD that underlie functional plasticity in ground state conformations, templated binding and folding, and a propensity to interact with several binding partners.

## Methods

**Protein sequences, expression, and purification.** The reconstruction of ancestral sequences and posterior probabilities of each amino acid residue were previously published[12]. Briefly, around 180 amino acid sequences of NCBD and CID domains, respectively, from present-day species were collected and aligned. The alignment and a species tree were used to reconstruct ancestral sequences using a maximum likelihood method. Of interest for the present paper, maximum likelihood (ML) ancestral sequences for two nodes were reconstructed: CID (and NCBD) from an early vertebrate at the time of the whole-genome duplications, denoted 1 R, and NCBD from the last common ancestor of deuterostome and protostome (bilaterian) animals, denoted D/P. AltAll versions of reconstructed sequences incorporate the second most likely amino acid in all uncertain positions, with a chosen cutoff value. The AltAll versions of $NCBD_{D/P}$ (Fig. 1) and $CID_{1R}$ used in the present paper was based on posterior probabilities from the reconstruction with a cutoff of 91%. One position in $NCBD_{D/P}$ (Thr2062) is very uncertain. A Thr residue (15%) was originally chosen at this position in $NCBD_{D/P}^{ML}$ to avoid a hydrophobic residue, although Ile (19%) has a higher posterior probability. In the current $NCBD_{D/P}^{AltAll}$ residue 2062 is Ile. NCBD and CID variants were expressed in *E. coli* from a pRSETA plasmid and purified as previously described[34]. Concentrations were determined by absorbance at 280 nm for NCBD or 205 nm for CID using calculated extinction coefficients based on amino acid composition. Purity was checked by SDS-PAGE and the identity of the purified proteins was confirmed by MALDI-TOF mass spectrometry.

**Circular dichroism.** Far-UV CD experiments were performed using a JASCO J-1500 CD spectrophotometer with a Peltier temperature control system. The protein concentration in all CD experiments was 18–20 µM. The buffer solution was either 20 mM sodium phosphate (pH 7.4), 150 mM NaCl or 50 mM potassium formate (pH 3.0), 150 mM NaCl, and CD spectra were recorded at 277 K. The scanning speed was 50 nm/min, bandpass 1 nm and integration time 1 s. Each reported spectrum is an average of 3–4 recorded spectra. In the equilibrium thermal denaturation experiments, the proteins were dissolved in 50 mM potassium formate buffer (pH 3.0), 150 mM NaCl, and the sample was heated from 277 to 368 K. The heating rate was 1 K/min, waiting time 5 s and the denaturation was monitored at 222 nm. During the heating, spectra (average of three measurements) were taken at 283, 303, 323, 343, and 363 K, respectively. In the equilibrium urea denaturation experiments the proteins were denatured in 0–6.6 M urea at 277 K and the unfolding was monitored at 222 nm. The data were analyzed in GraphPad Prism and fitted by nonlinear regression to a two-state equilibrium model[43]. Because of the broad unfolding transition and lack of a native baseline for several variants, four parameters were shared among all urea denaturation data sets in the curve fitting (Fig. S3 and Table S2). One of these parameters was the molar ellipticity of the native state, with the reasonable assumption that the secondary structures of the native states are similar for all variants. The other parameters were the slopes of the denatured and native baselines as well as the $m_{D-N}$ value (assuming a similar change in the solvent accessible surface area upon denaturation for all variants).

**Temperature-jump experiments.** Temperature-jump experiments were conducted using 100–200 µM protein on a TJ-64 temperature-jump system (TgK Scientific). The buffer solution was 20 mM sodium phosphate (pH 7.4), 150 mM NaCl. The proteins were subjected to temperature jumps of 2–8.5 K to different target temperatures. The data were fitted in the Kinetic Studio software (TgK scientific) to a double exponential function, to account both for the fast phase (relaxation time ∼80 µs), which corresponds to the heating time of the instrument, and to the slow phase, which corresponds to the conformational transition in the protein. A Trp variant of $NCBD_{D/P}^{ML}$ ($NCBD_{D/P}^{T2073W}$) was chosen for kinetic experiments since a Trp in this position was previously used for human NCBD. We engineered a Trp at other positions in $NCBD_{D/P}^{ML}$ and obtained rate constants in the same range for all variants (Table S2).

**Stopped-flow experiments.** The stopped-flow experiments were performed using an upgraded SX-17MV stopped-flow spectrophotometer (Applied Photophysics) and the measurements were conducted at 277 K. Proteins were dissolved in 5 mM HCl (pH ∼2) and rapidly mixed 1:1 with buffer solutions complemented with trimethylamine *N*-oxide (TMAO) to a final concentration of 20 mM sodium phosphate (pH 7.4), 150 mM NaCl, 1 M TMAO. The final protein concentration was 5 µM. The excitation wavelength was 280 nm and the resulting fluorescence emission was detected after passage through a 330 nm band-pass filter. The kinetic transients (1000 data points) were typically an average of 3–6 individual

acquisitions and were analyzed in GraphPad Prism by fitting to a single or double exponential function.

**ITC experiments.** The ITC experiments were performed using a MicroCal $ITC_{200}$ System in 20 mM sodium phosphate buffer (pH 7.4), 150 mM NaCl at 298 K. The different CID variants (concentrations in the syringe varying from 300–500 µM) were titrated to a 10-fold lower NCBD concentration in the cell, and the data were fitted to a two-state binding model to obtain estimates of the equilibrium dissociation constant ($K_D$), enthalpy of binding ($\Delta H$), and stoichiometry ($n$).

**NMR experiments.** NMR experiments were acquired on Bruker 600, 700, and 900 MHz spectrometers equipped with triple resonance cryogenic temperature probes at 298 K except otherwise stated. The final NMR samples contained 500 µM protein to which 0.01% $NaN_3$ and 5% $D_2O$ was added. Experiments for assignment and subsequent structure determination were done at pH 2.4 (HCl adjusted with NaOH) for $NCBD_{D/P}^{ML}$ and pH 6.8 for $NCBD_{D/P}^{AltAll}$. When we compared the 2D HSQC spectra of both proteins at these two pH values (2.4 and 6.8, respectively), we observed that most of the residues have similar chemical shifts. In addition, the CD spectra of both proteins at these pH values also indicate that they are folded to the same extent (Fig. 4). The reason for using the lower pH for $NCBD_{D/P}^{ML}$ is because, in addition to the mono-dispersed peaks around 8 ppm, additional well-separated peaks appear above 8 ppm, which enabled 3D-type assignment. $NCBD_{D/P}^{AltAll}$ on the other hand, showed a mono-dispersed spectrum at pH 2.4, but the spectrum at pH 6.8 was superior to that at pH 2.4 since most signals were intact and as such, 3D assignment was possible for data collected at pH 6.8. The following NMR experiments were recorded for assignment and subsequent structure determination: standard 2D $^1H$-$^{15}N$ HSQC, 3D HNCACB $^{15}N$-resolved [$^1H$-$^1H$]-NOESY-HSQC, $^{15}N$- resolved [$^1H$-$^1H$]-TOCSY-HSQC $^{13}C$-resolved [$^1H$-$^1H$]-NOESY-HSQC, HCCH-TOCSY, and $^{13}C$-resolved [$^1H$-$^1H$]-TOCSY-HSQC. Phi-angle restraining $^3J_{HNHA}$ couplings were determined from 3D HNHA type experiments using quantitative -*J* intensity-modulated experiments[44]. The temperature variation experiments were performed in 20 mM sodium formate, pH 3.0. All experiments were processed with nmrpipe[45] and analysed with CCPnmr[46].

**Structure calculation.** Structure calculations were done using the CYANA 3.97[47] package as follows: Initially, cross-peaks were converted into upper distance restraints following an automated process in CYANA. These distance restraints together with φ/ψ dihedral angles determined from $C^α$-chemical shifts and $^3J_{HNH}α$ (measured) were used as input for the initial structure calculations. The structures were calculated with 200,000 torsion angle dynamics steps for 100 conformers starting from random torsion angles by simulated annealing. Twenty conformers with the lowest target function values were selected and analyzed. The structural statistics together with all input data for the structure calculations are presented in Table 1. In the figures, the structures with the lowest CYANA target function are displayed, since these were similar to ensemble average structures. Structures with the lowest target functions were used in all analyses. Assignments and structural coordinates have been deposited in the RCSB protein data bank with PDB code: 7OSR (accession number BMRB ID 34635) ($NCBD_{D/P}^{ML}$) and 7OSW (accession

## Table 1 NMR distance and dihedral restraints.

| | $NCBD_{D/P}^{ML}$ | $NCBD_{D/P}^{AltAlt}$ |
|---|---|---|
| Distance restraints | | |
| Total NOEs | 329 | 347 |
| Intra-residue | 113 | 145 |
| Sequential ($|i - j| = 1$) | 122 | 111 |
| Medium-range ($1 < |i - j| < 4$) | 76 | 76 |
| Long-range ($|i - j| > 5$) | 18 | 39 |
| Total dihedral angle restraints | 76 | 63 |
| $^3J_{HN}α$ scalar couplings | 39 | 43 |
| $^{13}C^α$ chemical shifts | 100 | 100 |
| Structure statistics | | |
| Average CYANA target function value ($Å^2$) | 2.5 ± 0.2 | 2.8 ± 0.3 |
| Violations | | |
| Distance constraints (>0.5 Å) | 0 | 0 |
| Dihedral angle constraints (>5°) | 0 | 0 |
| Deviations from idealized geometry | | |
| Bond lengths (Å) | 0 | 0 |
| Bond angles (°) | 0 | 0 |
| Impropers (°) | 0 | 0 |
| Average pairwise r.m.s. deviation (Å) | | |
| Backbone residues | 2.2 ± 1.0 | 0.5 ± 0.2 |
| Heavy atom residues | 2.7 ± 1.0 | 1.0 ± 0.3 |

number BMRB ID 34636) (NCBD$_{D/P}$$^{AltAll}$). The solvent accessible surface area was calculated using the software ProtSA with a 1.4 Å solvent radius[48,49].

**Statistics and reproducibility**. The reproducibility of the type of biophysical experiments reported in this paper is generally high, as described in the reporting summary. All data were presented in the paper or in supplementary files.

**Reporting summary**. Further information on research design is available in the Nature Research Reporting Summary linked to this article.

## Data availability

All clones are available upon request. Assignments and structural coordinates have been deposited in the RCSB protein data bank with PDB code: 7OSR (accession number BMRB ID 34635) (NCBD$_{D/P}$$^{ML}$) and 7OSW (accession number BMRB ID 34636) (NCBD$_{D/P}$$^{AltAll}$). Source data for all graphs can be found in Supplementary Excel file 1. All other data are provided in the paper.

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

## Acknowledgements

This work was supported by the Wenner-Gren foundation WG-17 returning grants to C.N.C., by the Swedish Research Council (2020-04395 to P.J.), and Knut and Alice Wallenberg foundation (Evolution of new genes and proteins, to P.J.). F.A.S was funded by the Erasmus+ programme of the European Union. We used the NMR Uppsala infrastructure, which is funded by the Department of Chemistry—BMC and the

Disciplinary Domain of Medicine and Pharmacy. We thank Magnus Kjaergaard for his valuable comments on the NMR experiments.

## Author contributions

P.J. and C.N.C. conceived the project. E.K., P.J., and C.N.C. designed the experiments and wrote the paper. E.K., F.A.S., E.A., and C.N.C. performed experiments. All authors analyzed data. P.J. supervised the project.

## Funding

## Competing interests

The authors declare no competing interests.
