## [Peer Review File · Communications Biology]

Reviewers' comments:

Reviewer #1 (Remarks to the Author):

Karlsson et al. used ancestral reconstruction to construct an ancient sequence a ~50 residues region from the CREB binding protein, NCBD. They applied NMR, CD, stopped-flow fluorescence to demonstrate that NCBD is dynamic in both the ancient and extant versions but the extant one has increased stability.

I have many concerns about this manuscript, and I am hesitant to recommend its publication.

Majors are:

The NCBD has been evolved early. This disordered region might be used for interacting with other binding partners (e.g. CID) using the folding-upon-binding mechanism. I agree with the conclusion that the ancient and extant NCBD are both dynamic, this is probably the need to retain this property to interact with its partners. However, the interpretation from the stability is over-interpret. Because the ancient NCBD is live in a different organism, in a different environment. However, the experiments are comparing ancient and extend proteins in the same buffer condition. Maybe in another scenario, in the ancient condition, e.g. lower pH? the ancient NCBD is more stable, and during the evolution, it is less stale to have higher efficiency to bind its partner.

NCBD itself is an IDR, as the author states and many others have reported (in the references). The NCBD undergoes folding-upon-binding when it encounters CID. However, the authors solved this IDP's "folded" structure at the first beginning by a "serendipity" finding that this NMR signal is good at pH 2.4 (fig 1). The following discussion is to compare to extend NCBD at pH 6.8 (line 93-97). The conditions are very different and what is the meaning of comparing these two? I also wonder, later in the manuscript, the author can get reasonable NMR spectra at physiological pH (fig 4a). Why not solve the structure at the physiological condition at the first begging? The pH seems an important issue in this manuscript. I think the authors should indicate them more clearly. For example, what's the pH they used to solve the AltAll version (fig S1)? What is the pH for the NCBD/CID complex (fig. 2)? What is the pH for figures 3 and 5? Without a clear indication, it is difficult to verify the conclusion that the authors want to draw from the results.

In figure 2, does the ancient NCBD binds to the ancient CID or extant CID? There is no clue from the figure legends and the text. There is a CID(1R) used later, and if it is not comparing (ancient NCBD: ancient CID), (extant NCBD: extend CID), that is very strange. Again, what's the pH of the interacting proteins?

It is good to show the ITC's raw data (the binding titration) instead of bars, to be more convincing (Fig S2a)

The statements for figure 4, around lines 160-165: The sharpness of NMR HSQC spectra is not only affected by conformational heterogeneity but the exchanging kinetics. At slow or intermediate exchange regimes would have different results. Furthermore, the H/D exchange of amide protein is different depends on pH. This would also affect the line width in the HSQC. The interpretation of the NMR signal being sharpened is because of structural heterogeneity towards a single state at lower pH is over-interpretation. In addition to the thermodynamic factors, kinetics is also important.

Line 182-183: as mention above, the ancient NCBD sequence shows less deltaG than the extant one does not mean that the native states of the NCBD are less populated in a Cambrian animal because the proteins environments are different. Again, NCBD is an IDP, but from the text, it seems the folded state is defined as its "native" state? It is confusing.

Line 196: What do the authors mean by "a typical native to highly-disordered state transition"? The temperature was 363K. Does this protein can return to its original state at 283 K (from CD spectra) after visiting 363 K? If not, then the proteins were "cooked", and there is no need to discuss the native to disordered transition.

Line 198: Why do “well-dispersed peaks” at 333 K indicate a “globular, collapsed” structure? And in Fig S3, the 333 K signals (red) are less dispersed comparing to the other temperature. -

Although all variants introduce in the same position, the introduced fluorescence probe T2073W 2- to 3- fold enhances the protein stability (table S1). It may not be just “around 1 kcal/mol” (line 221). The author might have to provide other support to declare that the kinetics results are not affected by this replacement. As shown in Table S2, T2073W shows very different k_{obs} (two- to three-fold)

Line 236: The stopped-flow experiments were jumped from pH 2 to 7.4. Is this protein folded at acid pH (Figure 1)?

Minor

Some places are “AltAlt” but in figures, they are “AltAll”. please clarify.

An extra red dot at the bottom of figure 4c.

Sentences such as “we conclude that our conclusions are robust....(line 150)”, “the observed difference in stability between appears to be a robust result” (line 168) or “we can therefore conclude with greater confidence” should be avoided. The data will speak for themselves.

Reviewer #2 (Remarks to the Author):

The authors investigated the structural difference between two NCBD variants, the ancient version from 600 million years ago and the human version, using NMR. This is a very interesting idea to illustrate the necessity of the dynamic properties of NCBD, and the biophysics, sequence properties and evolutionary path controlling such structural property. The manuscript is well written. I recommend the manuscript to publish in Communications Biology and provide my minor comments below.

The authors used a phylogenetic method to obtain the ancient variant of NCBD. They did provide the citation of the method and a controlling sequence showing the worst case. However this is the starting point of the project, it would be great to show the details of the method in SI or maintext.

When comparing the structural properties between the three variants (human, ML and AltAll), it seems the RMSD between the human version and control is comparable to the RMSD between ML and the control. Then it's difficult to say whether the structural difference seen here is introduced by the evolution or by the method of reconstructing the ancient NCBD, though the other properties that were discussed later (e.g. binding affinity) seem to show the role of the control.

I am not sure if it's possible to provide a quantitative higher resolution structural comparison in addition to only RMSD. There are a few candidates: 1) angle between helix since the authors discussed tilt of helix; 2) the solvent accessible surface area of a specific region since the authors discussed the role of conserved hydrophobic core; 3) residue specific comparison experimentally or computationally. For instance R2/R1 and sPRE, which might be beyond the scope of the current work, could be useful to realize the dynamics or hydrophobic core.

It might be helpful to provide a summary table with all the variants and their properties: binding affinity, thermal, stability, pH and denaturant dependence, and structural properties such as tilt angle and SA of hydrophobic core.

It looks like when studying binding, CID sequence was also replaced with the ancient version. Do the authors have any comments regarding whether structural and functional improvement mainly comes from NCBD or CID? The sequence similarities between different variants including both

NCBD and CID might be helpful.

Considering the similar pH dependence, this might suggest the charged amino acids only contribute to nonnative instead of native on-pathway interactions and make the landscape more rough, whereas interactions between uncharged amino acids (e.g. hydrophobic core) contribute to folding. The authors also mentioned when reducing the pH, they could obtain a better NMR spectra, which is consistent with the picture that charged amino acids in NCBD might only provide nonnative interactions and introduce structural fuzziness. If the authors could quantify "better/worse" NMR spectra, they might be able to ask if evolution reduces/increases these nonnative interactions at neutral pH and makes the landscape more/less funnel-like.

Regarding the folding kinetics of the two variants, it looks like their solved structures do not differ that much. Does that mean a structure-based model will break down? Can a structure-based model still capture the different folding kinetics (two-state or three-state) and if the more complex transition state is realized in the aspect of the structure or difference in the sequence without altering that much the structure. This might be beyond the scope of the manuscript.

One other aspect interesting to ask in the future, what are the key mutations that improve the binding affinity and are they the same ones that impact the stability? Do these two types of mutations co-evolve with CID? Co-evolutionary analysis with CID might be helpful. It's expected that mutations altering binding affinity would co-evolve with CID, but if mutations that alter stability also co-evolve CID, then it might be an evidence that the structural properties of NCBD and flexibility of NCBD are important to its binding to CID.

Reviewer #3 (Remarks to the Author):

In the manuscript by Karlsson et al, the structure, thermodynamic stability and folding properties were measured for an ancestral reconstruction of Nuclear Coactivator Binding Domain (NCBD) from CBP. The ancestral reconstruction of the pre-Cambrian sequence has been previously described, with this work focusing on determining whether the molten-globule (MG) state of the extant human sequence also exists in these ancestral precursors. All data showed that reconstructed variants have very similar properties to the present-day version. This provides evidence that the dynamic properties of this domain may be a critical functional feature that has been evolutionarily conserved. This will be of interest for researchers interested in the role of the molten globule state in NCBD and more generally in protein function.

The characterization of reconstructed sequences was thorough and validated by comparison of the ancestor predicted by maximum likelihood and a variant of this reconstruction using second-choice substitutions at the six sites with lower certainty associated with the reconstruction. Both reconstructed sequences showed very similar behaviour, which supports the conclusion that the true ancestral proteins would also behave in this way. However, it should be noted that the majority of the substitutions appear at unstructured N- and C-terminal regions of the domain, and that there are no sequence changes in residues in the hydrophobic core. In addition, only 3 - 4 residues were mutated in the helical regions, the most consequential of which may be the mutation of residues at the N-terminus of helix 1. These mutations might be expected to affect the stability of the helix given the introduction of an additional proline residue in this region for both reconstructed sequences, although this was not noted. It would be informative to map the location of the mutations on the human NCBD structure in one of the figures.

Given the high level of sequence conservation in the regions of the domain responsible for creating the compact state, it is not surprising to see that a MG state persists in the ancestral reconstruction. While this might reflect evolutionary pressure to maintain the MG state, it should be acknowledged that the high sequence conservation is also dictated by its need to maintain interactions with a number of different binding partners.

When discussing the difference in helix orientations between the different variants, it should be noted that variability is to be expected for a MG that only shows a small number of long-range NOEs (less than one per residue) and that it can be problematic to represent a conformationally

dynamic ensemble with a single structure determined by conventional solution NMR methods. It would be useful to indicate whether there is any evidence to support the assumption that all NOEs arise from a single structure. In addition, while the data shows that the members of the ensemble being detected by the NOEs reflect compact structures, the absence of NOEs does not rule out the possibility that the same structures observed in other variants may also contribute to the ensemble but were not detected due to unfavourable dynamics. In general, the special considerations that come into play when determining a structure of a MG should be included in the discussion of the structures.

Minor changes:

Abstract: Capitalize Nuclear Coactivator Binding Domain

Page line 117 – The sentence is a little unclear. Recommend changing to “NCBDD/PAItAll differs from NCBDD/PML by amino acid substitutions at six positions.”

Reviewers' comments:

Reviewer #1 (Remarks to the Author):

Karlsson et al. used ancestral reconstruction to construct an ancient sequence a ~50 residues region from the CREB binding protein, NCBD. They applied NMR, CD, stopped-flow fluorescence to demonstrate that NCBD is dynamic in both the ancient and extant versions but the extant one has increased stability.

I have many concerns about this manuscript, and I am hesitant to recommend its publication.

Majors are:

The NCBD has been evolved early. This disordered region might be used for interacting with other binding partners (e.g. CID) using the folding-upon-binding mechanism. I agree with the conclusion that the ancient and extant NCBD are both dynamic, this is probably the need to retain this property to interact with its partners. However, the interpretation from the stability is over-interpret. Because the ancient NCBD is live in a different organism, in a different environment. However, the experiments are comparing ancient and extend proteins in the same buffer condition. Maybe in another scenario, in the ancient condition, e.g. lower pH? the ancient NCBD is more stable, and during the evolution, it is less stale to have higher efficiency to bind its partner.

Reply: Ancestral NCBD was reconstructed using sequences from extant deuterostome and protostome animals (Hultqvist et al., 2017, eLife). The ancestral creature was thus an animal, which lived around 600 million years ago in the sea. While we don't know the conditions within the cells of this animal, we note that all extant eukaryotes (including non-animals) have very similar cells. It is thus very likely that the cell biology of this ancestral animal is the same as in present day animals, including the pH value in different cellular compartments.

In any case, we don't make any strong interpretations from the difference in stability between ancestral and human NCBD, except noting that there is a difference, and this difference in thermodynamic stability is similar at low and high pH.

NCBD itself is an IDR, as the author states and many others have reported (in the references). The NCBD undergoes folding-upon-binding when it

encounters CID. However, the authors solved this IDP's "folded" structure at the first beginning by a "serendipity" finding that this NMR signal is good at pH 2.4 (Fig. 1). The following discussion is to compare to extend NCBD at pH 6.8 (line 93-97). The conditions are very different and what is the meaning of comparing these two?

Reply: First, we note that while NCBD is very dynamic, it has a hydrophobic core and folds into a rather well-defined 3D structure even in absence of a ligand (Kjaergaard et al, 2010, PNAS 28, 12535). This structure is overall retained upon interaction with CID.

And yes, the structures were determined at two different pH values. However, when we compared the 2D HSQC spectra of both proteins at these two pH values (2.4 and 6.8, respectively), we observed that most of the residues have similar chemical shifts. In addition, the CD signal of both proteins at these pH values also indicate that they are folded to the same extent (Fig. S3). The reason of using the lower pH for NCBD_ML is because, in addition to the mono-dispersed peaks around 8 ppm, additional well-separated peaks appear above 8 ppm, which made 3D-type assignment much easier. This assignment would not have been possible at pH 6.8. The human NCBD, on the other hand, showed a mono-dispersed spectrum at pH 2.4, but the spectrum at pH 6.8 was superior to that at pH 2.4 since most signals were intact and as such, a 3D assignment walk was possible. This has been clarified in the revised version of the manuscript.

I also wonder, later in the manuscript, the author can get reasonable NMR spectra at physiological pH (fig 4a). Why not solve the structure at the physiological condition at the first begging?

Reply: The reason of this is as stated above. The spectra at pH 2.4 gave a mono-dispersed spectrum (clearly visible around 8 ppm, 1H), additionally more distinct peaks were seen above 8 ppm (1H direction) which facilitated the 3D walk during assignment.

The pH seems an important issue in this manuscript. I think the authors should indicate them more clearly. For example, what's the pH they used to solve the AltAll version (fig S1)? What is the pH for the NCBD/CID complex (fig. 2)?

What is the pH for figures 3 and 5? Without a clear indication, it is difficult to verify the conclusion that the authors want to draw from the results.

Reply: We have provided information about pH in all figure legends in the revised version, and in Table S2. Note that pH does not affect overall thermodynamic stability much, but it could slightly shift conformational states.

In figure 2, does the ancient NCBD binds to the ancient CID or extant CID? There is no clue from the figure legends and the text. There is a CID(1R) used later, and if it is not comparing (ancient NCBD: ancient CID), (extant NCBD: extend CID), that is very strange. Again, what's the pH of the interacting proteins?

Reply: We have added information about CID and pH to the figure legends, and clarified what CID 1R is in the main text. We have published these structures earlier (Jemth, 2018, Sci Adv). The CID domain that interacts with NCBD appears to be a recent invention (ca 500 Myr) and the complex is between ancestral NCBD and an ancestral CID from around 440 Myr ago, as now stated in the text.

It is good to show the ITC's raw data (the binding titration) instead of bars, to be more convincible (Fig S2a)

Reply: We have added ITC data for three experiments to the figure.

The statements for figure 4, around lines 160-165: The sharpness of NMR HSQC spectra is not only affected by conformational heterogeneity but the exchanging kinetics. At slow or intermediate exchange regimes would have different results. Furthermore, the H/D exchange of amide protein is different depends on pH. This would also affect the line width in the HSQC. The interpretation of the NMR signal being sharpened is because of structural heterogeneity towards a single state at lower pH is over-interpretation. In addition to the thermodynamic factors, kinetics is also important.

Reply: Yes, the sharpness of the NMR HSQC spectra depends not only the conformational heterogeneity but also significantly on the rate of exchange between these conformational states. With respect to the NMR time regime, fast exchange will result in sharp peaks, intermediate exchange will result in broadening of the spectra while slow exchange will result in distinct conformational peaks. Thus, the exchange kinetics will definitely affect the rate of H/D exchange and the sharpness of the NMR lines. We have modified this sentence accordingly

Line 182-183: as mention above, the ancient NCBD sequence shows less deltaG than the extant one does not mean that the native states of the NCBD are less populated in a Cambrian animal because the proteins environments

are different. Again, NCBD is an IDP, but from the text, it seems the folded state is defined as its “native” state? It is confusing.

Reply: Regarding the ancestral conditions: As explained above, the conditions were likely very similar in the cells of the ancestral animal and today's animal cells. And even if they were not, we can still compare overall thermodynamic stability, in particular since the difference in thermodynamic stability is robust to changes in pH value. This finding is interesting in light of thermodynamic stabilities of other reconstructed proteins where the opposite is often observed, namely that the ancestral protein is more thermodynamically stable.

Regarding the native state: As noted above, NCBD is not a typical IDP but rather "molten-globule like" as described by Kjaergaard and Teilum (PNAS, 2010). NCBD has a hydrophobic core and populates a state, which can be solved by NMR. It is also clear that NCBD shows a cooperative (un)folding transition in urea-induced unfolding experiments. In this respect it is a folded domain and not an archetypical IDP. NCBD is however very dynamic, and, as pointed out in the introduction, these borderline molten-globule-like properties make NCBD particularly interesting for studies like the present one.

We describe these properties in the introduction:

The NCBD domain has a hydrophobic core, but is very dynamic and even verging on being intrinsically disordered (5–7). It has therefore been described as "molten-globule-like" (8–11).

Line 196: What do the authors mean by “a typical native to highly-disordered state transition”? The temperature was 363K. Does this protein can return to its original state at 283 K (from CD spectra) after visiting 363 K? If not, then the proteins were “cooked”, and there is no need to discuss the native to disordered transition.

Reply: The transition between denatured and native state(s) is reversible with temperature for NCBD as judged by CD experiments. It does not aggregate. We have added "reversible" to the following sentence (around line 190):
Previous experiments demonstrated that NCBD_{Human} shows an apparent non-cooperative reversible unfolding behavior when subjected to increasing temperatures, as monitored by CD (6, 8).

Line 198: Why do “well-dispersed peaks” at 333 K indicate a “globular, collapsed” structure? And in Fig S3, the 333 K signals (red) are less dispersed comparing to the other temperature. -

Reply: As noted above, the (un)folding process for NCBD is reversible with temperature and the protein returns to the folded state at 298 K after the experiments at 363 K. The spectrum at 333 K is not less dispersed than at other temperatures. For example, it has sharper peaks than that at 293 K.

The spectrum at 333 K is (well) mono-dispersed indicative of a single collapsed structure. In addition, distinct signals are seen around 8 ppm/ 115 ppm, suggesting that at 333 K most residues display a single NMR signal with sharp lines.

Although all variants introduce in the same position, the introduced fluorescence probe T2073W 2- to 3- fold enhances the protein stability (table S1). It may not be just “around 1 kcal/mol” (line 221). The author might have to provide other support to declare that the kinetics results are not affected by this replacement. As shown in Table S2, T2073W shows very different k_{obs} (two- to three-fold)

Reply: T2073W was chosen (among the variants in current Table S2) based on similarity to the wild type NCBD with regard to thermodynamic stability and CID binding (Karlsson et al., 2020, JBC, 295, 17698). In addition, previous folding experiments on human NCBD were performed using a Trp in the corresponding position (Dogan et al, 2016, Biochemistry). The mutation indeed stabilizes both ancestral and human NCBD and thus the rate constants of folding. We think a direct comparison between these two Trp variants is valid. We have changed the sentence to be more precise regarding the stabilization:

...resulted in a thermodynamic stabilization of 1.2-1.4 kcal mol⁻¹

After all, taking protein a domain out of its context in a full-length protein, and cellular conditions is always connected to some uncertainty, and so is labeling with in this case a Trp. But this is done all the time in biophysical experiments, including NMR, and we know that the information is usually useful when interpreted in a sensible way.

Line 236: The stopped-flow experiments were jumped from pH 2 to 7.4. Is this protein folded at acid pH (Figure 1)?

Reply: Yes, as shown in Fig. S3 the difference in thermodynamic stability between pH 3.0 and 7.4 is very small. So, what do we observe in these pH jumps? It is likely small population shifts between a denatured ensemble and two folded states, of which one state is dominating (the one obtained by NMR). This structural heterogeneity was previously investigated (Kjaergaard et al., 2013, Biochemistry; Dogan et al, 2016, Biochemistry) and the details are too complex to go into in the present paper. We explained this briefly in the kinetics section as a background to the present experiments, where we observe (un)folding kinetics of similar magnitude for the different variants.

Minor

Some places are “AltAlt” but in figures, they are “AltAll”. please clarify.

Reply: Fixed

An extra red dot at the bottom of figure 4c.

Reply: It has been removed.

Sentences such as “we conclude that our conclusions are robust.....(line 150)”, “the observed difference in stability between appears to be a robust result” (line 168) or “we can therefore conclude with greater confidence” should be avoided. The data will speak for themselves.

Reply: Line 150; Here we want to highlight that while reconstructed sequences may not be correct, our conclusions are still valid. We think this sentence explains this well, where "robust" refers to the conclusions and the uncertainty in sequence reconstruction.

Line 168, this sentence was deleted.

"with greater confidence" was also deleted.

Reviewer #2 (Remarks to the Author):

The authors investigated the structural difference between two NCBD variants, the ancient version from 600 million years ago and the human version, using NMR. This is a very interesting idea to illustrate the necessity of the dynamic properties of NCBD, and the biophysics, sequence properties and evolutionary path controlling such structural property. The manuscript is well written. I recommend the manuscript to publish in Communications Biology and provide my minor comments below.

The authors used a phylogenetic method to obtain the ancient variant of NCBD. They did provide the citation of the method and a controlling sequence showing the worst case. However this is the starting point of the project, it would be great to show the details of the method in SI or maintext.

Reply: The reconstruction of NCBD was published previously, and comprises extensive bioinformatics work (Hultqvist et al., 2017, eLife), which we think is not productive to recapitulate here. However, we have added a brief explanation of the reconstruction in the methods section.

When comparing the structural properties between the three variants (human, ML and AltAll), it seems the RMSD between the human version and control is comparable to the RMSD between ML and the control. Then it's difficult to say whether the structural difference seen here is introduced by the evolution or by the method of reconstructing the ancient NCBD, though the other properties that were discussed later (e.g. binding affinity) seem to show the role of the control.

I am not sure if it's possible to provide a quantitative higher resolution structural comparison in addition to only RMSD. There are a few candidates: 1) angle between helix since the authors discussed tilt of helix; 2) the solvent accessible surface area of a specific region since the authors discussed the role of conserved hydrophobic core; 3) residue specific comparison experimentally or computationally. For instance, R2/R1 and sPRE, which might be beyond the scope of the current work, could be useful to realize the dynamics or hydrophobic core.

Reply: These are good points and we followed the suggestions of the reviewer and have performed the following analyses using the lowest energy structures:

We measured the tilt angles between the first and last helices and report the results in updated figures.

We also calculated the SASA for folded and theoretically unfolded NCBD variants using the ProtSA software. The data are reported in a new supplementary table.

It might be helpful to provide a summary table with all the variants and their properties: binding affinity, thermal, stability, pH and denaturant dependence, and structural properties such as tilt angle and SA of hydrophobic core.

Reply: We have updated the supplementary tables. However, it is hard to have all data in one table.

It looks like when studying binding, CID sequence was also replaced with the ancient version. Do the authors have any comments regarding whether structural and functional improvement mainly comes from NCBD or CID? The sequence similarities between different variants including both NCBD and CID might be helpful.

Reply: Yes, we used an ancestral version of CID, CID_1R, which was present in the common ancestor of vertebrates. This variant was also reconstructed in the original work (Hultqvist et al., 2017, eLife) The first follow up study was the NMR characterization of ancestral and modern NCBD/CID complexes (Jemth et al, 2018, Sci Adv). The affinity of the oldest NCBD (the NCBD_D/P_ML used in the present work) was found to be lower than those of younger NCBD variants. We showed that this is the result of a complex interplay between both NCBD and CID. Thus, we addressed this question in the previous work. An alignment of NCBDs is shown in Fig. 1c.

Considering the similar pH dependence, this might suggest the charged amino acids only contribute to nonnative instead of native on-pathway interactions and make the landscape more rough, whereas interactions between uncharged amino acids (e.g. hydrophobic core) contribute to folding. The authors also mentioned when reducing the pH, they could obtain a better NMR spectra, which is consistent with the picture that charged amino acids in NCBD might only provide nonnative interactions and introduce structural fuzziness. If the authors could quantify "better/worse" NMR spectra, they might be able to ask if evolution reduces/increases these nonnative interactions at neutral pH and makes the landscape more/less funnel-like.

Reply: In the NMR timescales, the NCBD_Human spectrum is "better" at pH 6.8 compared to that at pH 2.4 while the NCBD_ML is better at pH 2.4 (more peaks appear sharper, likely due to fast exchange) compared to pH 6.8. This could mean that the NCBD_Human is less heterogeneous at higher pH, thus less fuzzy, while the opposite is true for the NCBD_ML. The only residues that could titrate between neutral and low pH are a His in NCBD_ML, a His in another position in NCBD_AltAll and an Asp in human NCBD (Fig. 1). Thus, all three variants likely gain one positive charge going from pH 7.5 to 3.0 and the similar small changes in stability is therefore not surprising. This, and the fact that many parameters affect the NMR spectra, we prefer not to draw conclusions regarding complex properties such as fuzziness.

Regarding the folding kinetics of the two variants, it looks like their solved structures do not differ that much. Does that mean a structure-based model will break down? Can a structure-based model still capture the different folding kinetics (two-state or three-state) and if the more complex transition state is realized in the aspect of the structure or difference in the sequence without altering that much the structure. This might be beyond the scope of the manuscript.

Reply: We are not sure that we're answering the question, but the conformational landscape of NCBD is complex, and yes, a full description is beyond the scope of this work. Data from previous NMR studies on human NCBD from Kjaergaard and Teilum (PNAS 2010), kinetics from us (Dogan et al., 2016, Biochemistry) and single molecule experiments from Ben Schuler's group (Nat Comm) suggest that NCBD exists in at least two folded conformations in addition to a denatured state, and where one is dominating (the NMR structure). The basis for the kinetic traces obtained from jumping from acid to neutral pH is likely a slight re-arrangement between such conformational states. We have added another sentence explaining this, since this phenomenon might also underlie the different qualities of NMR spectra at low and neutral pH.

One other aspect interesting to ask in the future, what are the key mutations that improve the binding affinity and are they the same ones that impact the stability? Do these two types of mutations co-evolve with CID? Co-evolutionary analysis with CID might be helpful. It's expected that mutations altering binding affinity would co-evolve with CID, but if mutations that alter stability also co-evolve CID, then it might be an evidence that the structural properties of NCBD and flexibility of NCBD are important to its binding to CID.

Reply: Yes, these are very interesting suggestions, and we have previously considered looking at co-evolution between NCBD and CID. However, this is complicated by several things. Firstly, NCBD has several protein ligands so any changes in the amino acid sequence must be compatible with all ligands, putting constraints on the sequence that are not directly related to binding of CID. (The flexibility of NCBD likely helps.) Secondly, there are two paralogs of NCBD (from CBP and p300, respectively) and three paralogs of CID, from NCOA1 (Src1), NCOA2 (Tif2) and NCOA3 (ACTR), respectively. Thus, there is likely a redundancy in function.

In previous data sets on other interaction domains where we plotted stability versus affinity we did not find any correlations. With NCBD, a very dynamic domain, there might be one. This is obviously a study in its own right but we agree it is worth looking into.

Reviewer #3 (Remarks to the Author):

In the manuscript by Karlsson et al, the structure, thermodynamic stability and folding properties were measured for an ancestral reconstruction of Nuclear Coactivator Binding Domain (NCBD) from CBP. The ancestral reconstruction of the pre-Cambrian sequence has been previously described, with this work focusing on determining whether the molten-globule (MG) state of the extant human sequence also exists in these ancestral precursors. All data showed that reconstructed variants have very similar properties to the

present-day version. This provides evidence that the dynamic properties of this domain may be a critical functional feature that has been evolutionarily conserved. This will be of interest for researchers interested in the role of the molten globule state in NCBD and more generally in protein function.

The characterization of reconstructed sequences was thorough and validated by comparison of the ancestor predicted by maximum likelihood and a variant of this reconstruction using second-choice substitutions at the six sites with lower certainty associated with the reconstruction. Both reconstructed sequences showed very similar behaviour, which supports the conclusion that the true ancestral proteins would also behave in this way. However, it should be noted that the majority of the substitutions appear at unstructured N- and C-terminal regions of the domain, and that there are no sequence changes in residues in the hydrophobic core. In addition, only 3 - 4 residues were mutated in the helical regions, the most consequential of which may be the mutation of residues at the N-terminus of helix 1. These mutations might be expected to affect the stability of the helix given the introduction of an additional proline residue in this region for both reconstructed sequences, although this was not noted. It would be informative to map the location of the mutations on the human NCBD structure in one of the figures.

Reply: Good suggestion, this has been done.

Given the high level of sequence conservation in the regions of the domain responsible for creating the compact state, it is not surprising to see that a MG state persists in the ancestral reconstruction. While this might reflect evolutionary pressure to maintain the MG state, it should be acknowledged that the high sequence conservation is also dictated by its need to maintain interactions with a number of different binding partners.

Reply: This is correct, at the end of the day, there will be selection for function. We have revised the discussion to put more emphasis on the multiple binding partners of NCBD. We speculate that the MG state is required to maintain all interactions.

When discussing the difference in helix orientations between the different variants, it should be noted that variability is to be expected for a MG that only shows a small number of long-range NOEs (less than one per residue) and that it can be problematic to represent a conformationally dynamic ensemble with a single structure determined by conventional solution NMR methods. It would be useful to indicate whether there is any evidence to support the assumption that all NOEs arise from a single structure.

Reply: We showed a single structure in the text for clarity. However, the complete ensemble is more heterogeneous. In addition, we had few long-range and many medium-range NOE distance restraints per residue, far fewer than the average for folded domains but enough to get a decent ensemble with well-defined backbone conformation. Thus, we showed the lowest energy structure which was not very different from the overall average. This has been explained in the revised version.

In addition, while the data shows that the members of the ensemble being detected by the NOEs reflect compact structures, the absence of NOEs does not rule out the possibility that the same structures observed in other variants may also contribute to the ensemble but were not detected due to unfavourable dynamics. In general, the special considerations that come into play when determining a structure of a MG should be included in the discussion of the structures.

Reply: Yes, the reviewer is right. The absence of NOEs does not rule out the possibility that the same structures observed in other variants may also contribute to the ensemble. This is the reason we looked for conditions in our NMR structure analysis that gave minimal conformational heterogeneity, thus enough NOE distance restraints. However, as it turns out not all residues were able to give favorable dynamics even at these conditions. We have included a discussion about this in the revised manuscript.

Minor changes:

Abstract: Capitalize Nuclear Coactivator Binding Domain

Page line 117 – The sentence is a little unclear. Recommend changing to “NCBDD/PAltAll differs from NCBDD/PML by amino acid substitutions at six positions.”

Reply: Done

REVIEWERS' COMMENTS:

Reviewer #1 (Remarks to the Author):

I thank Drs. Jemth and Chi's detailed reply. With their explanation in the response and additional words/figures/tables in the manuscript and supplementary materials, this article is much more comprehensive. It provides interesting insights into how intrinsically disordered proteins may have evolved to preserve their dynamic properties in interacting with their partners. I think it is in good form for publication.

Reviewer #2 (Remarks to the Author):

The authors have addressed all my previous questions in details. I recommend the manuscript to publish in Communications Biology. I have a minor comment. After obtaining more detailed structural insights, it looks like there can be some reconstruction errors which result in the structural variation between D/P and human NCBD. For instance for alpha-1, human variant is similar to AltAll; for alpha-3 human variant is close to ML; for SASA, human variant is close to AltAll. From these analysis, it might not be straightforward to tell how much the structure difference is not due to reconstruction. However there seems to be clear difference in folding thermodynamics between the ML/AltAll and human variants. Like the authors mentioned in the reply, NCBD landscape could be quite complicated and a conformational ensemble might be necessary to characterize the NMR data. Therefore these data including both NMR and thermodynamics of NCBD variants could be of great interest to the modeling community for further understanding the underlying structure driving force.

Reviewer #3 (Remarks to the Author):

The authors have done a good job of addressing reviewer comments, making the manuscript suitable for publication. There are some minor changes to the added text that I recommend, although these do not require re-review:

- 1) In the results it says the number of NOEs was "enough to get a reliable ensemble with well-defined backbone conformation." This is a bit misleading since a 'well-defined' conformation implies precision, which may be the case, but does not necessarily reflect the accuracy of that structure. My preference would be to say that the number of NOEs is sufficient to define the overall global fold.
- 2) I don't think this sentence needs to be in the main text: "The structures with the lowest CYANA target function were not different from the overall average and thus has been displayed in the main figures." It could be part of the figure legend, but should be changed to "Displayed here are structures with the lowest CYANA target functions since these were similar to ensemble average structures." Also include statement in Methods that structures with lowest target functions were used in all analyses/comparisons.
- 3) Rephrase: "However, as it turns out not all residues were able to give favorable dynamics under any conditions." to something like: "However it should be noted that there was still a small subset of residues that could not be resolved under any of the conditions tested, limiting the structural information that could be obtained from these regions." (Which residues? Would be useful to indicate this somewhere in Materials or Supplementary Material.)

Reviewer #2

The authors have addressed all my previous questions in details. I recommend the manuscript to publish in Communications Biology. I have a minor comment. After obtaining more detailed structural insights, it looks like there can be some reconstruction errors which result in the structural variation between D/P and human NCBD. For instance for alpha-1, human variant is similar to AltAll; for alpha-3 human variant is close to ML; for SASA, human variant is close to AltAll. From these analysis, it might not be straightforward to tell how much the structure difference is not due to reconstruction. However there seems to be clear difference in folding thermodynamics between the ML/AltAll and human variants. Like the authors mentioned in the reply, NCBD landscape could be quite complicated and a conformational ensemble might be necessary to characterize the NMR data. Therefore these data including both NMR and thermodynamics of NCBD variants could be of great interest to the modeling community for further understanding the underlying structure driving force.

Reply: We added one sentence to the paragraph below.

This result, which is consistent with the overall higher structural similarity between $NCBD_{D/P}^{AltAll}$, and $NCBD_{Human}$, may be due to the lack of NOE constraints as discussed above, but might also reflect the dynamic properties of NCBD, where small changes in sequence could modulate packing of helices. **In this respect, errors in the reconstruction could affect details of the structure.**

Reviewer #3

1) In the results it says the number of NOEs was “enough to get a reliable ensemble with well-defined backbone conformation.” This is a bit misleading since a ‘well-defined’ conformation implies precision, which may be the case, but does not necessarily reflect the accuracy of that structure. My preference would be to say that the number of NOEs is sufficient to define the overall global fold.

Reply: We have changed accordingly.

2) I don't think this sentence needs to be in the main text: “The structures with the lowest CYANA target function were not different from the overall average and thus has been displayed in the main figures.” It could be part of the figure legend, but should be changed to “Displayed here are structures with the lowest CYANA target functions since these were similar to ensemble average structures.” Also include statement in Methods that structures with lowest target functions were used in all analyses/comparisons.

Reply: Done

3)Rephrase: “However, as it turns out not all residues were able to give favorable dynamics under any conditions.” to something like: “However it should be noted that

there was still a small subset of residues that could not be resolved under any of the conditions tested, limiting the structural information that could be obtained from these regions.” (Which residues? Would be useful to indicate this somewhere in Materials or Supplementary Material.)

Reply: We have changed the sentence and listed the residues in the sentence: However, it should be noted that four residues (Ser2079, Gln2082, Ala2098, Gln2109) could not be resolved under any of the conditions tested, limiting the structural information that could be obtained from these regions.